# Effect of a Patient-Specific Structural Prior Mask on Electrical Impedance Tomography Image Reconstructions

**DOI:** 10.3390/s23094551

**Published:** 2023-05-07

**Authors:** Rongqing Chen, Sabine Krueger-Ziolek, Alberto Battistel, Stefan J. Rupitsch, Knut Moeller

**Affiliations:** 1Institute for Technical Medicine (ITeM), Hochschule Furtwangen, Jakob-Kienzle-Str. 17, 78054 Villingen-Schwenningen, Germany; 2Faculty of Engineering, University of Freiburg, Georges-Koehler-Allee 101, 79110 Freiburg, Germany

**Keywords:** electrical impedance tomography, structural prior, image reconstruction, inverse problem

## Abstract

Electrical Impedance Tomography (EIT) is a low-cost imaging method which reconstructs two-dimensional cross-sectional images, visualising the impedance change within the thorax. However, the reconstruction of an EIT image is an ill-posed inverse problem. In addition, blurring, anatomical alignment, and reconstruction artefacts can hinder the interpretation of EIT images. In this contribution, we introduce a patient-specific structural prior mask into the EIT reconstruction process, with the aim of improving image interpretability. Such a prior mask ensures that only conductivity changes within the lung regions are reconstructed. To evaluate the influence of the introduced structural prior mask, we conducted numerical simulations with two scopes in terms of their different ventilation statuses and varying atelectasis scales. Quantitative analysis, including the reconstruction error and figures of merit, was applied in the evaluation procedure. The results show that the morphological structures of the lungs introduced by the mask are preserved in the EIT reconstructions and the reconstruction artefacts are decreased, reducing the reconstruction error by 25.9% and 17.7%, respectively, in the two EIT algorithms included in this contribution. The use of the structural prior mask conclusively improves the interpretability of the EIT images, which could facilitate better diagnosis and decision-making in clinical settings.

## 1. Introduction

Electrical Impedance Tomography (EIT) is a medical imaging modality which is mainly used to visualise the ventilation distribution of the lungs at the bedside [1]. The estimated regional conductivity changes are calculated by electrical current injection and corresponding induced voltage measurements through electrodes placed equidistantly around the thorax [2,3]. Conductivity changes during respiration are mainly caused by the expansion of the alveoli, which lengthens the current pathways. One of the most common applications of EIT is monitoring the treatment of Acute Respiratory Distress Syndrome (ARDS) patients in the Intensive Care Unit (ICU) [4]. The resulting real-time insight into the ventilation distribution provided by EIT is valuable in adjusting ventilator settings, e.g., the Positive End-Expiratory Pressure (PEEP), or in changing the posture of mechanically ventilated patients [5,6]. Scientific and clinical research has shown that patients benefit from EIT, as it can reduce Ventilator-Induced Lung Injury (VILI) in mechanical ventilation [2,7]. One of the advantages of EIT is that it is radiation-free compared to other imaging methods, e.g., Computed Tomography (CT). Together with its relatively compact design and low-cost, EIT is suitable for frequent examinations or long-term monitoring. In addition, EIT is a real-time imaging modality that can reach 50 image frames per second.

Despite all of these benefits, EIT reconstruction leads to a mathematically ill-posed inverse problem characterized by large degrees of freedom, mainly because of the nonlinear nature of the relation between conductivity changes and boundary voltage measurements [8,9]. Reconstruction of an EIT image usually involves the use of the Finite Element Model (FEM) due to the inhomogeneity within the thorax. Several algorithms have been proposed to decrease the degrees of freedom of the EIT inverse problem, which modifies the FEM used for the inverse problem. Gong et al. used spectral graph wavelets to create a ’dual-model’ approach to the EIT inverse problem, resulting in down-sampling of the nodes in the FEM [10]. Schullcke et al. and Chen et al. published a new EIT algorithm that introduced the basic functions derived via Discrete Cosine Transformation (DCT) into the EIT image reconstruction [11,12,13]. The degrees of freedom of the EIT inverse problem become decreased through the clustering of the FEM by the cosine function subset from DCT. As computational power and parallel computing capabilities continue to improve, learning-based inverse solvers for fast EIT reconstruction are on the rise. One example of this is EIT image reconstruction based on structure-aware sparse Bayesian learning [14,15]. Deep Neural Networks are being used in EIT as well, with the deep D-bar algorithm enabling real-time reconstruction of absolute EIT images [16].

However, EIT images continue to be characterized by low spatial resolution, blurred anatomical alignment, and reconstruction-induced artefacts. These make them difficult to interpret in clinical settings. The combination of modalities as priors to improve image accuracy is a common concept, and can be applied to EIT. Several research groups have reported on the method of introducing structural prior information into the EIT reconstruction process [17,18,19,20]. Reconstructed EIT images showed improvements in interpretation; however, the priors used in these methods were static and not personalized. Patient-related structural priors have been reported by several other research groups as well. Nakanishi et al. established an anatomical atlas consisting of the probability distributions of tissue conductance obtained from measurements in multiple patients. However, this was not extended to personalization for other patients [21]. Zhang et al. used machine learning techniques to efficiently incorporate structural elements into the reconstruction process [22]; however, this is not computationally effective.

Additionally, it is common to incorporate data from one medical imaging technique into another in order to enhance the reconstruction or improve the interpretability of the results, e.g., PET-CT. There is a possibility of introducing patient-specific information obtained from, e.g., CT scans, into the EIT reconstruction process, such as through a structural prior mask [11]. The mask is capable of limiting the EIT reconstruction to the lung region, which provides a broader insight into the anatomical structure of the lungs. The applied structural prior could possibly achieve a superposition of the reconstructed conductivity change and morphological images. However, at present it is not clear how the EIT reconstruction is affected by an introduced structural prior mask.

The objective of this contribution is to investigate the influence of a personalized structural prior mask on EIT reconstructions, particularly using the Gauss–Newton (GN) EIT algorithm and DCT-based EIT algorithm. Two scopes of numerical simulations differing in terms of ventilation status and atelectasis scales were conducted. We reconstructed EIT images using the selected EIT algorithms with and without a structural prior mask. To analyse the influence of the structural prior mask, an evaluation of EIT reconstructions was conducted quantitatively in terms of the pixel-wise reconstruction error and figures of merit.

## 2. Methods

### 2.1. Structural Prior Mask in EIT Reconstruction Algorithm

An EIT algorithm reconstructs the variation of the electrical conductivity distribution $\hat{x}$ from the measurements of boundary voltages v; however, the reconstruction process in EIT is an ill-posed inverse problem. This means that the solution is not unique; additionally, an arbitrarily small perturbation in the boundary measurements can result in a large perturbation in the reconstruction of the conductivity variation. Time-difference EIT, which has been frequently used in clinical applications, is more resistant to perturbations unrelated to conductivity changes within the thorax, e.g., variations in electrode contact impedance, patient movements, and general measurement noise, even though the accurate conductivity distribution variation $x=\sigma_{2}-\sigma_{1}$ within the thorax is not linearly related to the boundary measurements of the induced voltage changes $y=v_{2}-v_{1}$. Our focus in this section is on the mathematical theory of EIT reconstruction using the Gauss–Newton (GN) EIT algorithm as an example, particularly the one-step linear Gauss–Newton solver. This approach is widely used in time-difference EIT applications, and is relevant to our discussion of established time-difference EIT algorithms in clinical settings. The reconstruction process is usually described as a minimization approach to estimate changes in the conductivity $\hat{x}$ from a set of changes in boundary voltages y, as follows:(1)$$\hat{x}={\mathrm{argmin}}_{x}\left\{{\left\|F\left(x\right)-y\right\|}_{2}^{2}+\lambda^{2}{\left\|\mathrm{Rx}\right\|}_{2}^{2}\right\}\;,$$
where $\hat{x}$ is the estimated conductivity change from the EIT reconstruction and F(x) represents the nonlinear model mapping the conductivity change x to the boundary voltage measurement y; y is usually normalized as $y=\frac{v_{2}-v_{1}}{v_{1}}$ in the time-difference EIT, while R is a regularization parameter introduced to linearize the inverse problem and λ is a hyperparameter that controls the level of regularization. There are different well-investigated choices of R, e.g., Tikhonov, NOSER, and Laplace. The choice of R can lead to different effects in the solution $\hat{x}$, such as preserving edges or smoothing neighboring impedance changes. As the conductivity properties of the thorax tissue are not homogeneous, an FEM is required to spatially discretize the domain.

In EIT, the conductivity changes are usually assumed to be small, smooth, and slowly varying. With this assumption, the forward model F(x) can be linearized around a preselected conductivity distribution reference $\sigma^{ref}$ as F(x)≈Jx. The Jacobian matrix J is a mapping from the voltage variations to the conductivity change; J derives from a background reference $\sigma^{ref}$ as $J_{i,j}={\left.\frac{\partial y_{i}}{\partial x_{j}}\right|}_{\sigma^{ref}}$, where the element $J_{i,j}$ maps small voltage changes at position *i* of y to a conductivity change of the element *j* within the FEM. With these assumptions, (Equation 1) can be solved in a linearized form:(2)$$\hat{x}={\left(J^{T}J+\lambda^{2}R\right)}^{-1}J^{T}y=\mathrm{By}\;,$$
where the matrix B is the reconstruction matrix. This represents the broadly used one-step linear Gauss–Newton solver.

To improve interpretability, a structural prior mask P can be derived from the morphological image H, e.g., CT or MRI, as follows:(3)$$P_{i,j}=\left\{\begin{array}{cc} 1, & \mathrm{if}\;H_{i,j}\in \mathrm{lung}\\ 0, & \mathrm{otherwise} \end{array}\right.\;.$$

Thereafter, the structural prior mask P can be integrated into the EIT reconstruction process by applying P to the corresponding space of the FEM as S=T(P). The operator *T* is a map used to assign every pixel in P to the element in the FEM model, which covers the pixel. The Jacobian matrix J is modified by the matrix S, which includes the structural prior mask, as $J_{mask}=J\cdot S$. Substituting the Jacobian matrix J in (Equation 2), the solution ${\hat{x}}_{mask}$ is calculated as
(4)$${\hat{x}}_{mask}={\left(J_{mask}^{T}J_{mask}+\lambda^{2}R\right)}^{-1}J_{mask}^{T}y=B_{mask}y\;.$$

In addition, the degrees of freedom in the EIT inverse problem can be reduced through the introduction of a subset of functions [11]. The subset is used to cluster the elements of the FEM, which introduces a map between the voltage variations and the changes of subset of the functions, i.e., the solution of the EIT inverse problem is represented by the change of the functions. Afterwards, an EIT image can be recovered using the inverse calculation of the clustering process. The subset of functions is of varying nature, e.g., a subset of cosine functions derived from the DCT. In this contribution, we include the DCT-based EIT algorithm into the evaluation, in which the solution of inverse problem is represented by the change of DCT coefficients. The details of this algorithm are described in Appendix A.

In this contribution, we chose the Tikhonov regularization as R=I. The optimal hyperparameter is chosen when the noise figure (NF) reaches 0.5. NF is defined as the ratio of the input signal-to-noise ratio (SNR) and the output signal-to-noise ratio
(5)$$NF=\frac{SNR_{in}}{SNR_{out}}=\frac{\mathrm{mean}|y|}{\sqrt{\mathrm{var}(n)}}/\frac{\mathrm{mean}|\mathrm{By}|}{\sqrt{\mathrm{var}(\mathrm{Bn})}}\;,$$
where n is the noise of the boundary voltage measurement.

### 2.2. Simulation Data

We evaluated the influence of the patient-specific structural prior mask on the EIT reconstructions by means of numerical simulations. The simulations were carried out with MATLAB R2019a (Mathworks, Natick, MA, USA) using the EIDORS toolbox [23]. A 3D FEM model for the numerical simulation experiments was generated by Netgen using the excursion of the thorax at the fifth intercostal space from a retrospective CT dataset [24]. For the initial setting of the simulation, i.e., the end of expiration, FEM elements which did not relate to the lung tissue were assigned a conductivity of $\sigma_{non-lung}^{initial}=1$, while elements relating to lung tissue were set to a conductivity of $\sigma_{lung}^{initial}=0.5$. The voltage measurement $v^{initial}$ was generated with the initial setting and used as the reference frame for the following EIT reconstruction. Boundary voltage measurements at end-inspiration were simulated in terms of two scopes of settings. The first scope of simulation settings was conducted with four different ventilation patterns:No ventilation in the dorsal right lungNo ventilation in the dorsal parts of both lungsNo ventilation in the ventral left lung and the dorsal right lungNo ventilation in the most ventral and most dorsal parts of both lungs.

As an example, ventilation pattern c is demonstrated in Figure 1. For the four different ventilation patterns, the FEM elements belonging to the non-ventilated area preserve the value of $\sigma_{lung}^{non-vent}=0.5$, while the conductivity of the FEM elements belonging to the ventilated lung area was changed to $\sigma_{lung}^{vent}=0.25$.

The second scope of simulation was designed as nine different increasing scales of atelectasis in the dorsal parts of both lungs, i.e., 10% to 90% with a step of 10%. These simulations are closer to a patient with a deteriorating status. Similar to the first simulation settings, the conductivity of the elements belonging to the ventilated lung region was changed to $\sigma_{lung}^{vent}=0.25$, while the non-ventilated area retained the value with $\sigma_{lung}^{non-vent}=0.5$.

For each of the simulation settings, the boundary voltage measurement $v_{i}^{vent}$ was generated accordingly. Additionally, we added 1% white noise to the boundary measurement $v_{i}^{vent}$ before the EIT reconstruction procedure. A different FEM mesh was implemented into reconstruction to prevent the ‘*inverse crime*’ [25]. It is worth noting that the structural prior masks integrated into the EIT reconstruction process were the binary lung contour, which was used to generate the lung area in the simulation models as well.

### 2.3. Evaluation of the Reconstruction

In this section, we introduce several evaluation parameters to quantitatively analyse the EIT reconstructions. These parameters are defined exclusively when the ground truth is known.

The *reconstruction error* (RE), which is calculated using the pixel-wise $\ell_{2}$-norm of the image differences between the reconstructed image and the ground truth, is the first evaluation parameter:(6)$$RE=∥H-H^{GT}{∥}_{2}\;,$$
where H is the EIT reconstruction and $H^{GT}$ represents the simulation ground truth, i.e., the simulated conductivity variations. Both H and $H^{GT}$ have *M* rows and *N* columns.

The figures of merit of the reconstructed image, which have been mentioned by several publications [10,19,26], were used for the evaluation. The following calculations are based on the definition of a lung region $H^{lung}$ derived from an EIT reconstruction H:(7)$$H_{i,j}^{lung}=\left\{\begin{array}{cc} 1, & \mathrm{if}\;H_{i,j}\leq 0.2\cdot H^{max}\\ 0, & \mathrm{otherwise} \end{array}\right.\;,$$
where $H^{max}$ represents the maximum pixel value in an EIT reconstruction H and $H^{lung}$ is a binary image that defines the lung region as ‘*one*’ and the non-lung region as ‘*zero*’. The lung region of the ground truth in the simulation is the same as P in (Equation 3). Illustrations of the definitions used in the calculation of the figures of merit are depicted in Figure 2.

*Dislocation* is the figure of merit used to evaluate the reconstructed position error of the lungs:(8)$$D=∥G_{l}^{GT}-G_{l}^{H}∥+∥G_{r}^{GT}-G_{r}^{H}∥\;,$$
where $G_{l}^{GT}$ represents the centre of gravity of the left lung in the ground truth $H^{GT}$, $G_{l}^{H}$ is the centre of gravity of the left lung in the reconstructed image H, $G_{r}^{GT}$ is the centre of gravity of the right lung in the ground truth $H^{GT}$, and $G_{r}^{H}$ is the centre of gravity of the right lung in the reconstructed image H. The left and right lung in reconstruction image are from the lung region $H^{lung}$.

*Shape deformation* quantitatively evaluates the corresponding lung regions $H^{lung}$ in the reconstruction H that do not comply with the actual area in the simulated ground truth $H^{GT}$:(9)$$SD=\frac{∥H^{lung}-P∥}{∥P∥}\;.$$

*Ring effect* is the ratio of the sum of the negative pixel values and the sum of the absolute pixel values in the reconstruction H:(10)$$R=\frac{\sum_{i,j}\left|H_{i,j}^{<0}\right|}{\sum_{i,j}\left|H_{i,j}\right|}\;,$$
where $H_{i,j}^{<0}$ represents the pixels with values less than zero in the EIT reconstruction H.

*Amplitude response* is defined as the ratio of the sum of the pixel values of the reconstruction H and the sum of the simulation ground truth $H^{GT}$:(11)$$A=\frac{\sum_{i,j}H_{i,j}}{\sum_{i,j}H_{i,j}^{GT}}\;.$$

In general, a small value for the figures of merit represents a good reconstruction. The only exception is the *amplitude response*. It should ideally stay constant, which allows for a quantitative analysis, e.g., air distribution within the lung area.

## 3. Results

The conductivity reconstructions for the four different ventilation patterns using the GN and DCT approaches are depicted in Figure 3. All the pixels of the images were normalised between −0.5 to 0.5, ensuring that all the images can be displayed with the same colormap. Black areas show no conductivity change, while blue or purple areas indicate the decrease or increase of the conductivity, respectively. The first row of Figure 3 demonstrates the conductivity changes applied in the simulation, i.e., the ground truth. The second row and third row illustrate the EIT reconstructions using the GN approach without and with a structural prior mask, respectively. The last two rows illustrate the reconstructions using the DCT approach without or with a structural prior mask, respectively. Without a structural prior mask, the EIT reconstructions lead to blurred edges with rather strong ring effects, which can be found in the reconstructions of both approaches (GN and DCT). In addition, anatomical details, e.g., the bronchus, with no conductivity change within the lung, cannot be observed in the reconstructions without a structural prior mask, while these details are preserved in the results using a structural prior mask. It is worth noting that even without a structural prior mask the EIT reconstructions from the DCT approach bear less artefacts.

The reconstruction errors calculated using the pixel-wise $\ell_{2}$-norm of the image difference between reconstructed images and the ground truth data are depicted in Figure 4. It is obvious that a structural prior mask implemented in both approaches, the GN and the DCT, will reduce the RE in the reconstructions when compared to the results without a structural prior mask. The RE is reduced by 25.9% and 17.7% on average in the GN and the DCT approach, respectively. There is no large difference between the results when using the GN and DCT approaches when the structural prior mask is applied.

The figures of merit of the reconstructions using the GN or DCT approaches with and without a structural prior mask are depicted in Figure 5. In general, introducing a structural prior mask to using the GN or DCT approach decreases the dislocation, shape deformation, and ring effect compared to the results without a structural prior mask. This complies with the finding in Figure 3, e.g., a reduced ring effect around the edge. In addition, even without a structural mask the amplitude response stays more stable in the DCT approach than in the GN approach.

The results for the simulations in terms of different atelectasis scales are illustrated in Figure 6. All the pixels in the images were normalised to between −0.5 to 0.5 for comparison purpose. Similar to results shown in Figure 3, the reconstructions with a structural prior implemented show a reduced ring effect. The anatomical structure and the edge of the lung are well preserved. As the atelectasis scale increases, the reconstructed ventilated areas become wider than in the simulated ground truth (cf. the blue areas in Figure 6, with 80% and 90% atelectasis scales). This is the widening effect of EIT, which occurs even when a structural prior mask is applied. In addition, conductivity changes are reconstructed at the dorsal part of the lung in the results from the GN approach with a structural prior mask, while in ground truth there are no conductivity changes. This might be due to the nature of the EIT inverse problem.

The corresponding reconstruction errors are depicted in Figure 7. Introducing a structural prior mask removes around 34.2% of the RE in EIT images using the GN approach and 23.9% using the DCT approach. Furthermore, as the atelectasis scale increases, i.e., as the ventilated lung area decreases, the RE should decrease as well. This is due to the reduction of the lung area with the changing conductivity. However, when no structural prior mask is applied in the GN approach and DCT approaches, the again RE increases after an atelectasis scale of 50% is reached. This might be due to the widening effect found in Figure 6. Similar to the finding in Figure 4, the influence of the structural prior mask is similar in both the GN and DCT approaches.

The figures of merit calculated from each EIT image in Figure 6 are listed in Table 1 and demonstrated as a box plot in Figure 8. It is clear that both the GN and DCT approaches with a structural prior mask produce more accurate reconstructions, which are represented by smaller values in terms of the dislocation, shape deformation, and ring effect. There are several outliers in Figure 8, which are mostly provided by the values calculated from the simulation with 80% and 90% atelectasis scales. The dislocation, shape deformation, and ring effect are the largest with these simulation settings, which confirms the findings in Figure 6. Figure 8 shows that the amplitude response remains more stable in the DCT approach with the structural prior mask, while it becomes much larger and is not ideally constant with the 80% atelectasis simulation setting.

## 4. Discussion

In this contribution, we implemented two scopes of numerical simulations to investigate the influence of a structural prior mask on EIT images using the GN and DCT approaches. In the first scope of simulations, four different ventilation patterns were simulated. A decreasing atelectasis scale of the dorsal lung was conducted in the second scope of our simulations. Overall, the EIT images reconstructed using the GN and DCT approaches benefited from the implemented structural prior thanks to improved interpretability. This result secures a more accurate EIT reconstruction as confirmed by different quantitative evaluations, including the reconstruction error and figures of merit.

The structural prior mask considerably improves the accuracy and interpretability of the resulting EIT image. In Figure 3 and Figure 6, the EIT images reconstructed with a structural prior mask generally maintained the shape of the lungs and preserved the anatomical details when using either the GN or DCT approach, making the overall EIT images much clearer and easier to interpret. This contributes to a decrease in the reconstruction error, dislocation, and shape deformation (Figure 5 and Figure 8). Ring effects in the non-lung regions caused by the reconstruction algorithm are prevented (compare the results without and with the structural prior mask presented in Figure 3 and Figure 6). The amplitude response stays rather stable when the structural prior mask is implemented, which could be helpful for calculating the regional air distribution within the lungs. In addition, introducing a structural prior mask into the EIT reconstruction process can facilitate the development of comprehensive insight into the pathophysiology of lungs in clinical settings, as the regional behaviour of the lungs can be directly correlated with the morphological structures.

It is common to combine data from different medical imaging methods to achieve better interpretability. However, it is worth noting that structural prior masks introduced to the EIT reconstruction process are usually derived from CT or MRI images representing the patient’s status at a certain time, while EIT is generally used to monitor a patient’s continuously changing status. In our simulation, the structural prior represents two healthy lungs without atelectasis, which means that the structural priors will not be accurate if there are atelectasis or non-ventilation areas in the simulation. Nevertheless, the majority of EIT images show benefits from the structural prior mask (cf. Figure 3 and Figure 6). However, when the difference between the current simulation setting and the structural prior mask becomes larger, e.g., 80% and 90% atelectasis in the simulation, the structural prior mask remains, as in healthy lungs the widening effect in the EIT images is obvious. The figures of merit confirm this finding as well. Even though the EIT images show improvements when compared to the results without a structural prior mask, the accuracy of the structural prior is crucial when introducing it into the EIT reconstruction process. An inaccurate structural prior mask, i.e., one that does not comply with the current patient status, can result in an off-base reconstruction. To avoid misleading EIT reconstructions and resulting compromised diagnosis, it is important to carefully select an accurate prior and detect any inaccuracies. Chen et al. have proposed a method for quantifying the error introduced by a structural prior mask, which can serve as an indicator of an inaccurate prior [12,13]. Consequently, it is necessary to check the implemented structural prior mask regularly to maintain its accuracy. Furthermore, the widening effect is a general problem in EIT which requires further investigation [27]. There are various additional sources of artefacts in EIT raw data, e.g., patient movement and electrode connection variability, that can produce a strong voltage signal. Without the implementation of the structural prior mask, the induced artefacts usually occur at the edge of the EIT image. There is a risk that these sources of artefacts may be reconstructed as the conductivity change within the lung region due to the introduction of the structural prior mask. However, there are methods to filter out components not correlated with the ventilation after reconstruction [28,29].

The fact that this study only used simulation experiments to evaluate the impact of the structural prior mask is one of its limitations. Additional studies with phantoms and clinical data are necessary to validate this method. In this contribution, we introduced the structural prior through the application of a corresponding mask. There are further methods to include patient-specific structural prior information into EIT algorithms, e.g., through sub-domain based regularization or soft-prior regularization [30,31]. Future studies should include these methods.

Nevertheless, reconstruction using the GN or DCT approaches can lead to enhanced interpretability if an accurate structural prior mask is implemented. The incorporation of structural information from a prior mask and functional data from EIT creates patient-specific EIT imaging. This enables a comprehensive understanding of pulmonary pathophysiology by preserving the anatomical structure of the lungs and correlating regional lung behavior with morphological structures.

## 5. Conclusions

Simulations with two different scopes were used to evaluate the impact of a structural prior mask on the EIT images from two different EIT algorithms. In conclusion, using a structural prior mask can improve the accuracy of EIT images. This is confirmed by the reduced RE and comparison using figures of merit. In addition, anomalies in ventilation and various conductivity distributions can be connected to respected structural data from other morphological imaging methods. This leads to better interpretability of the EIT images. Overall, the introduction of a structural prior mask represents a significant improvement over existing EIT algorithms, and has great potential for enhancing the precision of EIT in medical diagnostics.

## Figures and Tables

**Figure 1 sensors-23-04551-f001:**
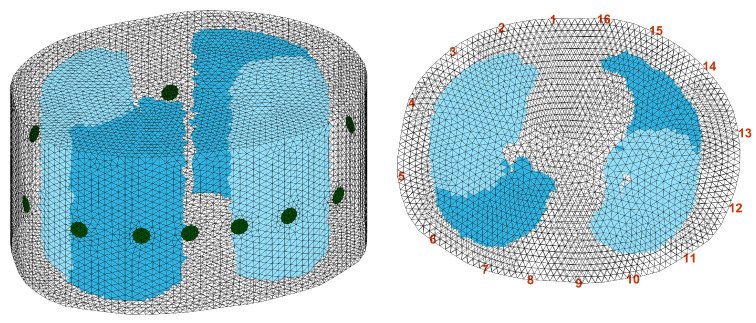
The 3D FEM model generated for simulation; simulation pattern c is demonstrated. The dark blue area represents $\sigma_{lung}^{non-vent}=0.5$, while the light blue area represents $\sigma_{lung}^{vent}=0.25$. **Left**: the 3D FEM model with simulation pattern c. **Right**: the aerial view of the 3D FEM model with simulation pattern c and the numbers of the electrodes.

**Figure 2 sensors-23-04551-f002:**
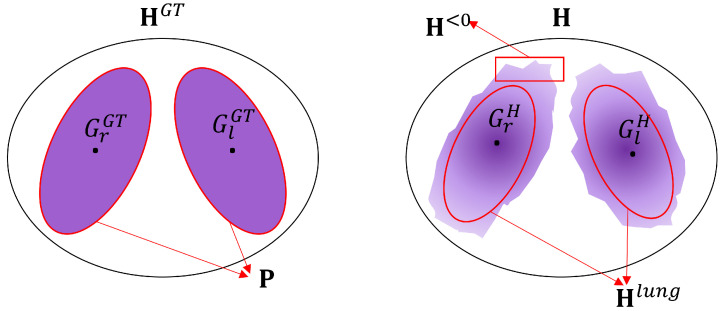
Illustrations of the definitions used in the calculation of the figures of merit. **Left**: Simulation ground truth $H^{GT}$. **Right**: EIT reconstruction H.

**Figure 3 sensors-23-04551-f003:**
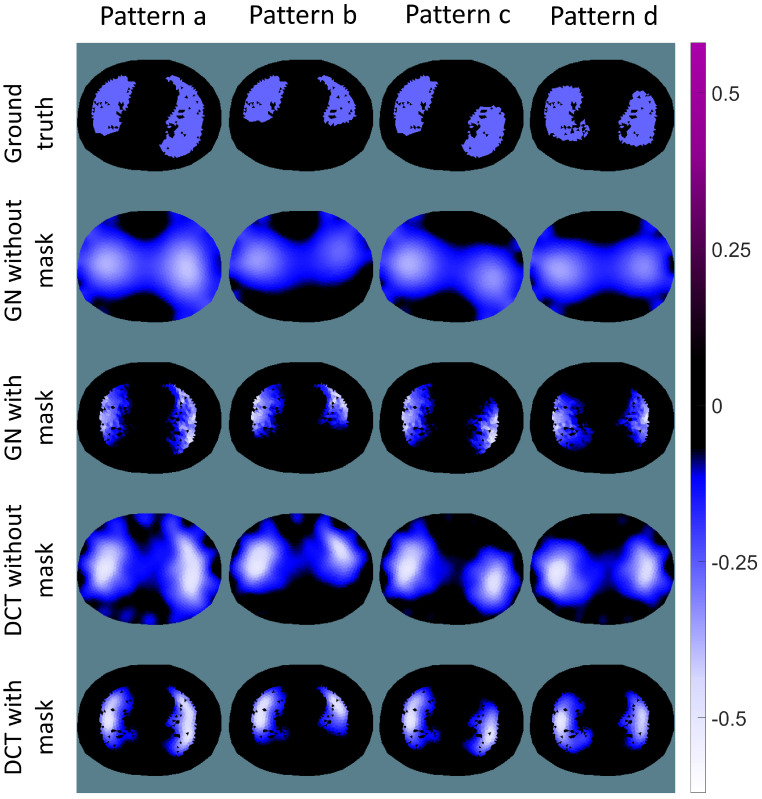
Ground truth and EIT reconstructions using the GN and DCT approaches. First row: different patterns of conductivity change for simulation (ground truth); second and third row: reconstructions of the conductivity change using the GN approach without or with a structural prior mask; fourth and fifth row: reconstructions of the conductivity change using the DCT approach without or with a structural prior mask.

**Figure 4 sensors-23-04551-f004:**
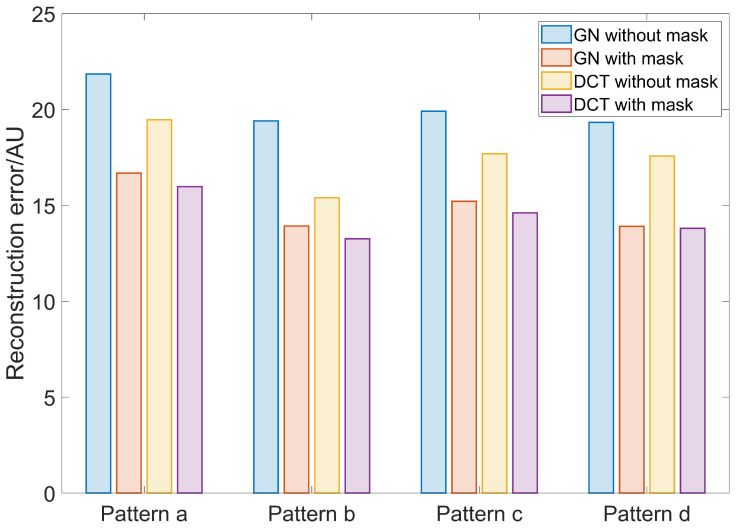
The reconstruction error of the results using the GN and DCT approaches without and with a structural prior mask. Blue bars: GN without structural prior mask; orange bars: GN with structural prior mask; yellow bars: DCT without structural prior mask; purple bars: DCT with structural prior mask.

**Figure 5 sensors-23-04551-f005:**
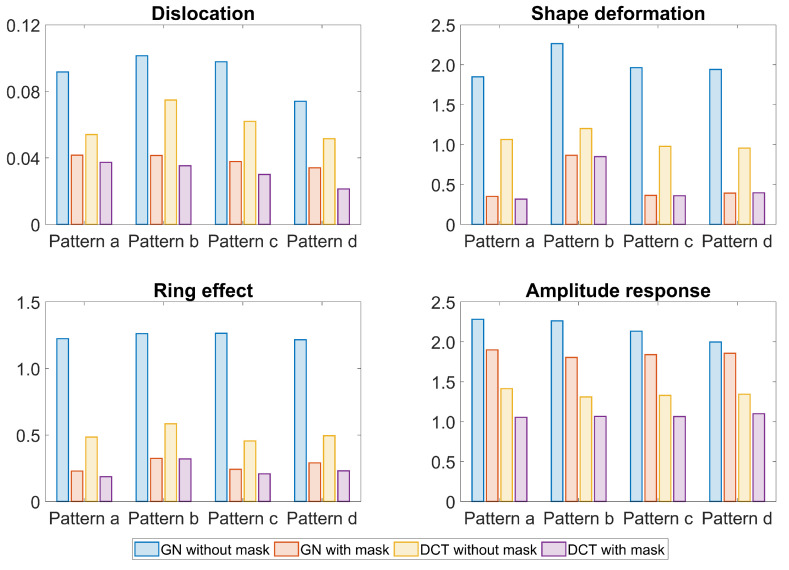
The figures of merit of the reconstructions using the GN and DCT approaches without or with a structural prior mask. Blue bars: GN without structural prior mask; orange bars: GN with structural prior mask; yellow bars: DCT without structural prior mask; purple bars: DCT with structural prior mask.

**Figure 6 sensors-23-04551-f006:**
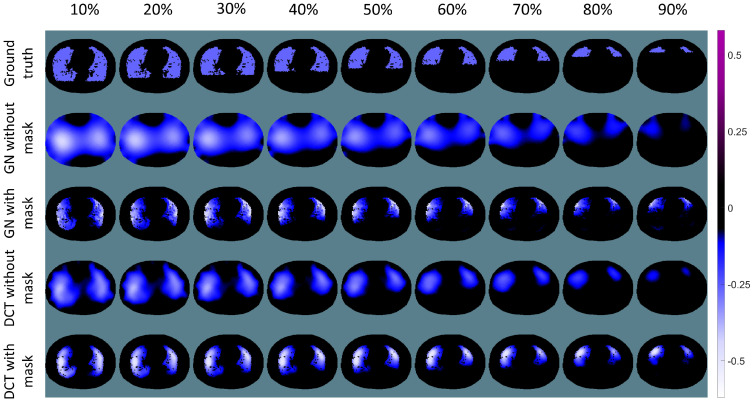
Ground truth and the EIT reconstructions of different scales of atelectasis. First row: ground truth in the simulation of different scales of atelectasis; second and third rows: reconstructions of the conductivity change using the GN approach without or with a structural prior mask; fourth and fifth rows: reconstructions of the conductivity change using the DCT approach without or with a structural prior mask.

**Figure 7 sensors-23-04551-f007:**
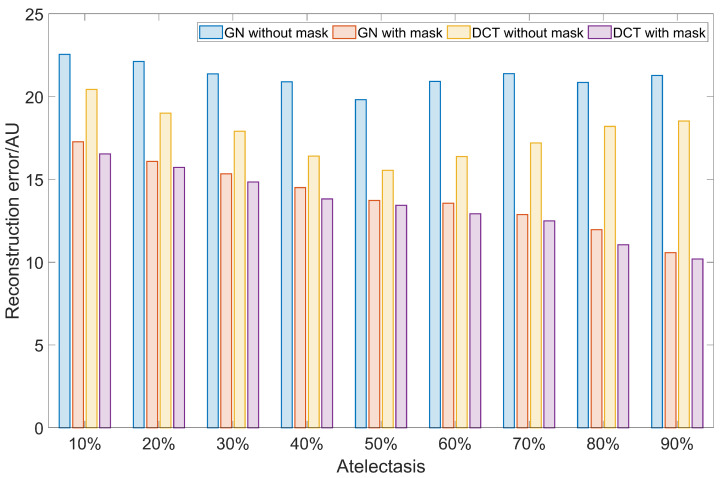
The reconstruction error calculated from the results of the simulation in terms of different scales of atelectasis. Blue bars: GN without structural prior mask; orange bars: GN with structural prior mask; yellow bars: DCT without structural prior mask; purple bars: DCT with structural prior mask.

**Figure 8 sensors-23-04551-f008:**
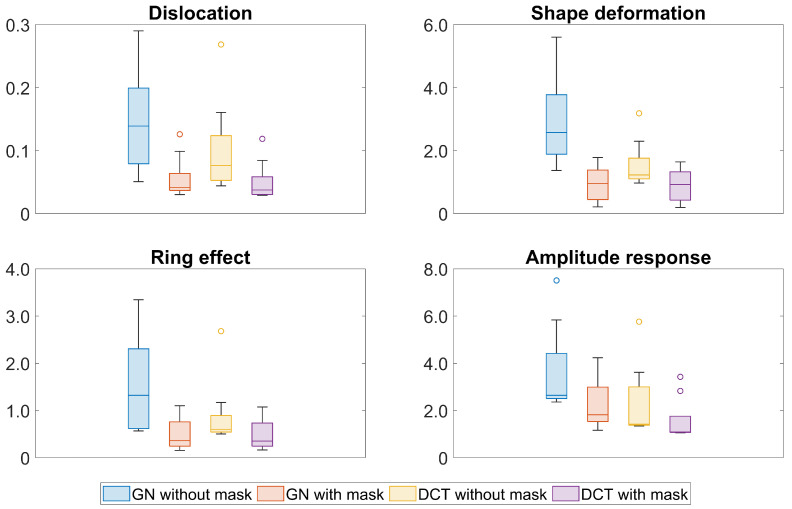
The figures of merit calculated from the EIT reconstructions of the simulation in terms of different scales of atelectasis. Blue box: GN without structural prior mask; orange box: GN with structural prior mask; yellow box: DCT without structural prior mask; purple box: DCT with structural prior mask.

**Table 1 sensors-23-04551-t001:** The figures of merit calculated from each EIT image of the simulation in terms of different scales of atelectasis.

Algorithm		GN								DCT							
Figures of Merit		D		SD		R		A		D		SD		R		A	
Mask		W/o	W	W/o	W	W/o	W	W/o	W	W/o	W	W/o	W	W/o	W	W/o	W
Scale	10%	0.051	0.030	1.368	0.214	0.566	0.153	2.548	1.159	0.044	0.029	0.969	0.194	0.503	0.162	1.418	1.084
	20%	0.061	0.035	1.692	0.341	0.586	0.219	2.564	1.547	0.049	0.030	1.019	0.314	0.519	0.214	1.380	1.070
	30%	0.085	0.037	1.948	0.479	0.627	0.251	2.371	1.475	0.054	0.030	1.130	0.466	0.551	0.252	1.370	1.054
	40%	0.113	0.040	2.134	0.712	0.729	0.286	2.640	1.761	0.065	0.033	1.227	0.669	0.586	0.286	1.374	1.064
	50%	0.139	0.041	2.573	0.960	1.320	0.361	2.359	1.818	0.076	0.037	1.226	0.922	0.601	0.352	1.343	1.069
	60%	0.152	0.044	2.804	1.075	1.894	0.553	2.722	2.303	0.090	0.040	1.274	1.044	0.653	0.501	1.641	1.173
	70%	0.191	0.052	3.446	1.312	2.143	0.712	3.943	2.837	0.112	0.050	1.583	1.282	0.802	0.681	2.788	1.400
	80%	0.224	0.099	4.751	1.587	2.791	0.900	5.831	3.440	0.160	0.085	2.296	1.454	1.170	0.897	3.614	2.826
	90%	0.290	0.126	5.600	1.779	3.343	1.099	7.504	4.236	0.268	0.119	3.184	1.639	2.679	1.071	5.758	3.423

D: Dislocation; SD: Shape deformation; R: Ring effect; AR: Amplitude response; W/o: Without mask; W: With mask.

## Data Availability

Not applicable.

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
