# Peer review of "Effect of a Patient-Specific Structural Prior Mask on Electrical Impedance Tomography Image Reconstructions"

_sensors, 2023, doi:10.3390/s23094551_

Round 1
Reviewer 1 Report
In this manuscript, the authors have introduced a patient-specific structural prior mask into EIT reconstruction process, aiming at the improvement of the image interpretability.The results demonstrate that the use of the structural prior mask improves the interpretability of the EIT images. Overall, the study is novel and well conducted, and the manuscript is clearly written, and has practical importance in medical diagnostics. So I recommend accepting it for publication.
Author Response
Thank you for your time and effort in reviewing our manuscript. We are pleased that you found our study to be novel and well conducted, and that you recognize the practical importance of our work in medical diagnostics.
Reviewer 2 Report
It is an interesting paper.
The authors applied the structural prior to the EIT imaging problem of a lung area.
What I understood as the methodology from this paper is the following. (1) This work limits the inverted solution space only to lung area and update the conductivity parameters only in the lung area. (2) At the same time, this work does not update any conductivity parameters outside lung area.
It is true that, if we reduce the solution space, we can somewhat alleviate the solution multiplicity of an inverse problem and reduce the inversion error between the reconstructed solution and the ground truth counterpart. However, such benefit necessitates that the information of the conductivity outside the lung area should be accurate. Otherwise, the inaccuracy of the conductivity outside the lung area will be detrimental to the inversion performance.
I think the authors have not described about this aspect. It will be highly recommended for them to elaborate the effect of the inaccurate conductivity information outside the lung area on the inversion performance of their presented method and how to address the detrimental effect.
Author Response
Thank you for your review of our manuscript. We understand your concern about the accuracy of conductivity information outside of the lung area and the potential impact of this on the inversion performance of our method. We agree that this is an important issue that needs to be addressed. We would like to clarify that all prior information implemented to EIT algorithms suffer from this problem, because the conductivity outside of the lung area is usually assumed to be homogeneous. In addition, the changes in conductivity within the non-lung area are usually negligible compared to the changes in air content within the lung area.
We acknowledge that inaccurate conductivity information can have an impact on the reconstructed images. In fact, we have conducted relevant research on this topic and developed a parameter called the "redistribution index" to detect inaccurate conductivity assigned to the priors. When the prior is deviating too much and may lead to misleading results, the redistribution index will reveal that special care is required and further steps such as prior update or replacement may be indicated [1], [2].
We have added further discussion on this concern in the revised manuscript:
[Page 11]…To avoid misleading EIT reconstructions and compromising the diagnosis, it is important to carefully select an accurate prior and detect any inaccuracies. Chen et al. have proposed a method to quantify the error introduced by a structural prior mask, which can serve as an indicator of an inaccurate prior [1], [2].
Reference:
[1] R. Chen and K. Moeller, ‘Redistribution Index – Detection of an Outdated Prior Information in the Discrete Cosine Transformation-based EIT Algorithm’, in 2021 43rd Annual International Conference of the IEEE Engineering in Medicine Biology Society (EMBC), Nov. 2021, pp. 3693–3696. doi: 10.1109/EMBC46164.2021.9630567.
[2] R. Chen, S. J. Rupitsch, and K. Moeller, ‘Influence of hyperparameter on the Untrue Prior Detection in Discrete Transformation-based EIT Algorithm’, in 2022 44th Annual International Conference of the IEEE Engineering in Medicine & Biology Society (EMBC), Jul. 2022, pp. 580–583. doi: 10.1109/EMBC48229.2022.9871293.
Reviewer 3 Report
The article presents the influence of a patient-specific structural mask on impedance tomography image reconstructions.
Minor remarks:
1) What is the advantage of the proposed solution over other methods used in solving the inverse problem in the EIT?
2) The authors could more broadly refer to other learning methods and justify the choice of the presented solution.
3) What is new and innovative about the article?
Author Response
1) What is the advantage of the proposed solution over other methods used in solving the inverse problem in the EIT?
Thank you for your valuable feedback. While it is true that EIT has been used to display ventilation distribution among patients, its low spatial resolution has made it difficult to interpret EIT reconstructions from most of the algorithms. To obtain a more comprehensive insight into pulmonary pathophysiology, it is necessary to combine structural and functional data, as is done in SPECT-CT or PET-CT. In our study, we addressed this issue by incorporating a CT-derived structural prior mask into the EIT reconstruction process. Compared to other algorithms, our approach not only effectively prevented the appearance of artefacts in non-lung regions but also preserved the anatomical structure of the lungs, resulting in images that the changes of the conductivity can be directly superimposed with high-resolution morphological data. This facilitates a more comprehensive insight into the pathophysiology of the lungs, as regional information about functional status can be directly correlated with the morphology of the lungs. Overall, our approach represents a significant improvement over existing EIT algorithms, and we believe it has great potential for enhancing the accuracy and clinical usefulness of EIT in medical diagnostics.
We have revised our Discussion part to address this concern:
[Page 12]… By combining structural information from a prior mask and functional data from EIT, the resulting patient specific EIT imaging preserves the anatomical structure of the lungs and prevents artifacts in non-lung regions. This enables a comprehensive understanding of pulmonary pathophysiology by preserving the anatomical structure of the lungs and correlating regional lung behavior with morphological structures.
[Page 12]…Overall, the introduction of the structural prior mask represents a significant improvement over existing EIT algorithms and has great potential for enhancing the precision of EIT in medical diagnostics.
2) The authors could more broadly refer to other learning methods and justify the choice of the presented solution.
Thank you for your suggestion. We agree that recent advances in learning methods for EIT, such as Deep D-bar and structure-aware sparse Bayesian learning, are promising and worth mentioning in our manuscript. However, we would like to clarify that the focus of our paper is on the comparison with established time difference EIT algorithms in clinical settings. As such, we have chosen to highlight the regularization algorithms that are widely used in EIT, including the classical Gauss-Newton solver, which represents most of the structurally similar algorithms for EIT reconstruction. We also note that these algorithms have been widely applied in clinical EIT, including in the Dräger PulmoVista® 500, one of the few commercial EIT devices with proper approval for clinical use.
While we have not extensively compared our approach with other learning methods for EIT in our paper, we acknowledge that this is a rapidly evolving area of research, and we have included some of these methods in the introduction part of our revised manuscript to provide more insights into the EIT algorithms:
[Page 2]…As computational power and parallel computing capabilities continue to improve, learning-based inverse solvers for a fast EIT reconstruction are on the rise. One example is the EIT image reconstruction based on structure-aware sparse Bayesian learning [1], [2]. Deep Neural Networks are also being used in EIT, with the deep D-bar algorithm enabling real-time reconstruction of absolute EIT images [3].
[Page 3]…Our focus in this section is on the mathematical theory of EIT reconstruction using the Gauss-Newton (GN) EIT algorithm as an example, particularly the one-step linear Gauss-Newton solver. This approach is widely used in time-difference EIT applications, and is relevant to our discussion on established time difference EIT algorithms in clinical settings.
3) What is new and innovative about the article?
Thank you for your comment. We would like to further elaborate on the innovative aspect of our proposed approach. While it is common to combine imaging modalities, such as SPECT-CT or PET-CT, to gain a more comprehensive insight as the functional status can directly be correlated with the morphological structure. However, this has not been previously implemented in EIT despite its low spatial resolution. In our proposed approach, structural information from the prior mask and functional data from EIT are fused to allow easier orientation for clinicians. This fusion can be viewed as a 'patient specific EIT imaging'. This incorporation of prior knowledge about the lung structure can provide a more interpretable EIT reconstruction for clinical use. Regarding the clinical applicability, all what is needed is a patient specific CT image, which will be available in many cases. Thus, we think that introducing structural prior mask from CT to EIT is a practical way to obtain a more interpretable EIT reconstruction for clinical use.
We have revised our Discussion part to address the innovative:
[Page 12]…The incorporation of structural information from a prior mask and functional data from EIT creates patient specific EIT imaging. This enables a comprehensive understanding of pulmonary pathophysiology by preserving the anatomical structure of the lungs and correlating regional lung behavior with morphological structures.
Reference:
[1] S. Liu, J. Jia, Y. D. Zhang, and Y. Yang, ‘Image Reconstruction in Electrical Impedance Tomography Based on Structure-Aware Sparse Bayesian Learning’, IEEE Trans. Med. Imaging, vol. 37, no. 9, pp. 2090–2102, Sep. 2018, doi: 10.1109/TMI.2018.2816739.
[2] S. Liu, H. Wu, Y. Huang, Y. Yang, and J. Jia, ‘Accelerated Structure-Aware Sparse Bayesian Learning for Three-Dimensional Electrical Impedance Tomography’, IEEE Trans. Ind. Inform., vol. 15, no. 9, pp. 5033–5041, Sep. 2019, doi: 10.1109/TII.2019.2895469.
[3] S. J. Hamilton and A. Hauptmann, ‘Deep D-bar: Real time Electrical Impedance Tomography Imaging with Deep Neural Networks’, IEEE Trans. Med. Imaging, vol. 37, no. 10, pp. 2367–2377, Oct. 2018, doi: 10.1109/TMI.2018.2828303.